# Pilot Study on Predictive Traits of Fresh Maize Hybrids for Estimating Milk and Biogas Production

**Radko Loučka [1], Filip Jančík [1,*], Petr Homolka [1], Yvona Tyrolová [1], Petra Kubelková [1], Alena Výborná [1], Veronika Koukolová [1], Václav Jambor [2], Jan Nedělník [3], Jaroslav Lang [3] and Marie Gaislerová [1]**

[1] Institute of Animal Science, Přátelství 815, 104 00 Prague, Czech Republic; loucka.radko@vuzv.cz (R.L.); homolka.petr@vuzv.cz (P.H.); tyrolova.yvona@vuzv.cz (Y.T.); kubelkova.petra@vuzv.cz (P.K.); vyborna.alena@vuzv.cz (A.V.); koukolova.veronika@vuzv.cz (V.K.); koukolova.marie@vuzv.cz (M.G.)

[2] NutriVet Ltd., Vídeňská 1023, 691 23 Pohorelice, Czech Republic; jambor.vaclav@nutrivet.cz

[3] Agricultural Research, Ltd. Troubsko, Zahradní 1, 664 41 Troubsko, Czech Republic; nedelnik@vupt.cz (J.N.); lang@vupt.cz (J.L.)

\* Correspondence: jancik.filip@vuzv.cz; Tel.: +420-267-009-650

**Abstract:** Farmers need information on which maize hybrid is best and under what conditions. They demand that this information be clear, simple and easily understood. This study aims to estimate the potential for milk production (MPP) and the biochemical methane potential (BMP) production from fresh maize hybrids. Using these indicators from fresh maize, information on the differences between hybrids can be effectively obtained, albeit with some of the shortcomings of this proposed method. Samples of fresh maize plants ($n = 384$) from four hybrids were evaluated at two locations over four consecutive years (from 2018 to 2021). The dry matter content, averaged across all hybrids, all years and both locations, was $371 \pm 42.3$ g.kg$^{-1}$. The colder and wetter the year, the significantly higher the starch content, lower the amylase-treated neutral detergent fibre content (aNDF) and lower the crude protein (CP), which was reflected in lower BMP. Weather did not significantly affect the net energy of lactation (NEL) or MPP values. The location significantly affected all monitored indicators, except BMP. The earlier the hybrid was at harvest time, the lower the NEL and MPP but the higher BMP contents were. This study is preliminary and must be repeated with more hybrids and under more different conditions.

**Keywords:** forage species; ruminant nutrition; digestibility; milk production potential; methane

## 1. Introduction

Maize (*Zea mays* L.) is the world's staple food crop for both humans and animals, especially ruminants. Maize silage is one of the most important ingredients used in dairy cattle diets, but it is characterised by a great variability in both nutrient composition and fermentation quality [1].

Maize has stable nutritional properties and high energy values, and it is relatively easy to grow and store. However, there are major differences between hybrids [2]. In addition to chemical composition, maize hybrids often differ in their amylase-treated neutral detergent fibre (aNDF) and aNDF digestibility (aNDFD); it is clear that aNDFD is a very important parameter for maize used as whole-crop silage for dairy cows [3]. The possibility of estimating silage energy and milk yield for the evaluation of maize hybrids according to aNDFD was pointed out [4]. Values of aNDFD change dramatically during the growing season, so it is very important to determine which growth phase has the best values [5]. The importance of the aNDFD of whole maize plants for dairy cow milk production has been highlighted [6,7]. The aNDFD values affect not only the NEL but also the MPP defined by [8]. Recently, knowledge of BMP has become increasingly important [9–11].

Drought is one of the most common negative factors facing maize in terms of growth, development and production. Further knowledge of drought tolerance could help maize

production [12]. Some maize hybrids are more tolerant to drought, depending on the vegetative growth stage of the plant [13]. It is not only the climatic conditions that are important but also, for example, the characteristics of the growing location [14] and several other factors.

Fresh maize as a material for the study was chosen with the knowledge of a certain error in comparison with silage (dry matter losses occur during fermentation). However, farmers are mainly interested in the differences between hybrids and, respectively, differences between factors (i.e., which hybrid is best). Absolute values are less important in this case. It is also important to note that experiments with silage or even animals are difficult and expensive. The evaluation of maize hybrids according to MILK 2006 in fresh plants was used, for example, by Mussadiq et al. [15] or Moreno-Resendez et al. [16]. The evaluation of maize as a substrate for biogas production has been discussed in detail by Weißbach [17], Weißbach [18] and Amon et al. [19].

This study aimed to evaluate selected predictive traits of fresh maize hybrids based on chemical analyses using a multivariate approach to estimate their use in silage production for ruminant feeding, and possibly for methane production.

## 2. Materials and Methods

A total of 4 maize hybrids of maturity class 240–280 FAO (Food and Agriculture Organization) were tested. Their characteristics are shown in Table 1. The experimental hybrids were selected from the maize hybrid portfolio with respect to their FAO earliness group (early or intermediate), grain endosperm type (semi-flint or flint), maturation type (uniformly maturing or with a tendency to stay green), and recommended use (for silage and/or for methane production). In a period of four consecutive years (from 2018 to 2021), hybrids were grown at two locations: L1 = Prague (50.0365000 N, 14.6090225 E) and L2 = Troubsko (49.1727358 N, 16.4981503 E). The distance between them is about 200 km, and both are located in a beet production area. Their altitude is about 280 m above sea level, and in both locations, the soil in the experimental fields was classified as loamy to clayey with neutral soil reaction.

**Table 1.** Basic characteristics of tested hybrids.

| Items | Hybrid | | | |
| | H1 | H2 | H3 | H4 |
|---|---|---|---|---|
| Mark | DKC3872 | DKC3450 | DKC3568 | DKC3575 |
| FAO | 240–250 | 250 | 260–270 | 270–280 |
| Maturity | early | early | turn to medium | turn to medium |
| Endosperm | turn to flint | semi-flint | semi-flint | turn to flint |
| Senescence | turn to SG | turn to SG | NSG | turn to SG |
| Use for | silage | SiloEnergy | silage | SiloEnergy |

FAO = number indicates earliness; SG = stay-green; NSG = standard senescence; SiloEnergy = intended not only for silage, but also for biogas production.

At both locations, the hybrids were sown on approximately the same dates in mid-April. The experiment was based on randomised blocks with four replicates per hybrid. The crop density was 95,000 plants per hectare. The pre-crop (wheat), tillage method, fertilisers and plant protection products used were almost identical in both locations. Data on plant development and uniformity were recorded throughout the growing season. The values of these parameters varied within the normal range, and there were no significant differences between the two experimental sites. Harvesting took place at the beginning of September, and grain silage maturity was monitored continuously. The experimental plots were harvested when the plants of a given hybrid had reached the two-thirds milk stage of the grain line. During the harvest, three samples from each field block were taken for laboratory analysis in full compliance with Commission Regulation (EC) [20].

Detailed data on the development of weather in both localities in the vegetation period from 2018 to 2021 are shown in Table 2 (average temperatures) and Table 3 (sum of precipitation).

**Table 2.** Average temperatures (°C) in the growing season 2018–2021 (source: CHMI 2021).

| Items | Year | IV. | V. | VI. | VII. | VIII. | IX. | Average |
|-------|------|-----|-----|-----|------|-------|-----|---------|
| L1 | 2018 | 13.3 | 16.9 | 18.2 | 20.8 | 21.5 | 15.3 | 17.7 |
| | 2019 | 10.0 | 11.4 | 21.5 | 19.8 | 19.5 | 14.1 | 16.1 |
| | 2020 | 10.1 | 11.7 | 17.0 | 18.7 | 19.6 | 14.8 | 15.3 |
| | 2021 | 6.2 | 11.1 | 19.5 | 19.1 | 16.7 | 15.1 | 14.6 |
| | 1981–2010 | 8.1 | 13 | 16.3 | 17.8 | 17.2 | 13.6 | 14.3 |
| L2 | 2018 | 14.5 | 18.1 | 19.6 | 21.6 | 22.8 | 16.0 | 18.8 |
| | 2019 | 11.1 | 12.2 | 22.0 | 20.4 | 20.8 | 16.6 | 17.2 |
| | 2020 | 9.9 | 12.6 | 18.0 | 19.5 | 20.6 | 15.0 | 15.9 |
| | 2021 | 7.1 | 12.7 | 20.2 | 20.8 | 17.9 | 15.6 | 15.7 |
| | 1981–2010 | 8.7 | 13.8 | 16.9 | 18.3 | 17.8 | 14.1 | 14.9 |

L1 = locality Prague; L2 = locality Troubsko.

**Table 3.** Precipitation (mm) in the growing season 2018–2021 (source: CHMI 2021).

| Items | Year | IV. | V. | VI. | VII. | VIII. | IX. | Sum |
|-------|------|-----|-----|-----|------|-------|-----|-----|
| L1 | 2018 | 19 | 54 | 69 | 27 | 33 | 49 | 251 |
| | 2019 | 25 | 72 | 47 | 52 | 72 | 46 | 314 |
| | 2020 | 21 | 64 | 120 | 40 | 99 | 64 | 408 |
| | 2021 | 23 | 102 | 95 | 107 | 84 | 16 | 427 |
| | 1981–2010 | 43 | 63 | 70 | 82 | 75 | 47 | 380 |
| L2 | 2018 | 11 | 36 | 36 | 40 | 15 | 86 | 137 |
| | 2019 | 17 | 22 | 65 | 60 | 56 | 73 | 221 |
| | 2020 | 20 | 65 | 87 | 59 | 106 | 82 | 338 |
| | 2021 | 16 | 58 | 67 | 100 | 130 | 15 | 372 |
| | 1981–2010 | 33 | 64 | 71 | 63 | 65 | 39 | 296 |

L1 = locality Prague; L2 = locality Troubsko.

In the laboratory, the chopped feed samples (cuttings) were first dried at 55 °C for 48 h. The dry samples were then homogenised in a laboratory grinder with a 2 mm mesh (for the in sacco method) and a 1 mm mesh (for laboratory analysis). The homogenised samples were analysed using standard methods according to AOAC [21], and the following contents were determined according to the method of Van Soest et al. [22]: DM, starch, CP, ash and aNDF. The digestibility of nutrients (aNDFD and OMD) was determined using the in sacco method in the rumen of Holstein cows with a cannula according to Ørskov and McDonald [23], expressed as a percentage of the nutrient ingested: $100 \times$ (weight—residue)/weight. Cows (dry cows) were fed meadow hay (ad libitum) with 2 kg of supplemental mix. Samples were incubated for 24 h. The NEL was calculated according to the Van Es method [24], adapted by Vencl et al. [25], where the following equation was used: NEL = (0.463 + (0.24 × (ME/BE))) × ME; where ME = (0.01549 × (OMD × 10)) and BE = (0.00588 × CP + 0.01918 × OM). The MPP (in litres per tonne of DM) was calculated according to MILK 2006 [8]. For calculation, aNDFD was added to the software after incubation for 24 h. Truly digested starch (IS-IV) was used. The BMP values were measured according to the guidelines for the VDI 4630 standard [26] using the method described by Amon et al. [19]. The experiments were performed on a device with a 48-nest set of glass anaerobic fermenters (reactors) heated to a mesophilic temperature of 38–40 °C. Mixing was ensured automatically using a timer (15 min every hour). An eudiometer batch reactor type, with the volume of individual vessels being 4 L, was used. The resulting

biogas exited from each fermenter into an immersion pressureless gas meter with a volume of 6 L, which also served for short-term biogas collection for subsequent measurement.

Analysis of variance (ANOVA) with multivariate design was used [27]. The statistical model for the results was $Y_{ijkl} = \mu + H_i + Y_j + L_k + HY_{ij} + HL_{ik} + YL_{jk} + HYL_{ijk} + e_{ijkl}$, where $Y_{ijkl}$ is the dependent variable, $\mu$ is the overall mean, $H_i$ is the effect of hybrid ($i$ = 1 to 4), $Y_j$ is the effect of year ($j$ = 1 to 4), $L_k$ is the effect of location ($k$ = 1 to 2), $HY_{ij}$ is the interaction of hybrid with year, $HL_{ik}$ is the interaction of hybrid with location, $YL_{jk\,k}$ is the interaction of year with location, $HYL_{ijk}$ is the interaction of hybrid with year with location and $e_{ijkl}$ is an error term. Tukey's HSD (honestly significant difference) test at a significance level $p < 0.05$ was used to evaluate the results. The associations for every item among factors (years, locations and hybrids) and also among characteristics of hybrids (maturity, endosperm, senescence and use) were evaluated using a bivariate correlation analysis. The probability of correlation (*p*-value) was calculated, and Pearson bivariate correlations were considered significant at $p < 0.001$ and $p < 0.01$.

## 3. Results and Discussion

### 3.1. Overall Overview of Influences

A summary of statistical significance by hybrid, year of experiment and location is given in Table 4. There were significant differences between hybrids for all parameters. There were significant differences between years for all parameters except NEL and MPP. There were significant differences between sites for all indicators except CP and BMP. Interactions between H and Y were significant for chemical analyses and BMP, but not for aNDFD, NEL and MPP. The reverse was true for the interaction between H and L, i.e., they were significant for aNDFD, NEL and MPP, but not for chemical analyses and BMP. The interactions between Y and L were significant for all indicators.

**Table 4.** Statistical significance by hybrid (H), year (Y), location (L) and their interactions (*).

| Items | H | Y | L | H*Y | H*L | Y*L | H*Y*L |
|---|---|---|---|---|---|---|---|
| DM g.kg$^{-1}$ | 0.008 | 0.000 | 0.017 | 0.000 | 0.451 | 0.000 | 0.001 |
| Starch g.kg$^{-1}$ DM | 0.000 | 0.000 | 0.030 | 0.036 | 0.983 | 0.000 | 0.044 |
| aNDF g.kg$^{-1}$ DM | 0.003 | 0.000 | 0.000 | 0.021 | 0.067 | 0.000 | 0.107 |
| CP g.kg$^{-1}$ DM | 0.005 | 0.000 | 0.565 | 0.046 | 0.964 | 0.000 | 0.444 |
| Ash g.kg$^{-1}$ DM | 0.001 | 0.000 | 0.000 | 0.043 | 0.758 | 0.000 | 0.103 |
| aNDFD % | 0.000 | 0.000 | 0.000 | 0.106 | 0.001 | 0.000 | 0.002 |
| NEL MJ.kg$^{-1}$ DM | 0.000 | 0.592 | 0.000 | 0.418 | 0.002 | 0.001 | 0.015 |
| MPP kg.t$^{-1}$ DM | 0.000 | 0.592 | 0.000 | 0.418 | 0.002 | 0.001 | 0.015 |
| BMP L.kg$^{-1}$ DM | 0.001 | 0.000 | 0.357 | 0.005 | 0.146 | 0.000 | 0.026 |

DM = dry matter; aNDF = amylase-treated neutral detergent fibre; CP = crude protein; aNDFD = aNDF digestibility; NEL = net energy of lactation; MPP = milk production prediction; BMP = biochemical methane potential; ± = standard deviation.

### 3.2. Estimation of Harvest Date, Weather and Location

The DM content, averaged across all hybrids, all years and both locations, was $371 \pm 42.3$ g.kg$^{-1}$. Comparing this with the literature recommendation [28] for the appropriate harvest time, harvest was slightly delayed. Bal et al. [28] determined that for maize, the optimum stage for silage production was at the two-thirds milk line stage of the grain (with some flexibility to vary between the one-quarter and two-thirds milk line), i.e., between 324 and 350 g.kg$^{-1}$ DM. Carpentier and Cabon [2] and Peyrat et al. [29] recommended a similar interface, namely DM content of 300 to 350 g.kg$^{-1}$. Maize was only harvested in this DM range in 2020 (Table 5). The higher DM content of whole maize plants (371 g.kg$^{-1}$) measured by us indicates harvesting in two-thirds of the milk line, which corresponds to the achievement of average starch (294 g.kg$^{-1}$ DM) and aNDF (448 g.kg$^{-1}$ DM) content. Bal et al. [28] reported similar average values for starch (287 to 372 g.kg$^{-1}$ DM) and for aNDF (444 to 405 g.kg$^{-1}$ DM).

**Table 5.** Chemical composition and feed value of hybrids in different years.

| Items | 2018 | 2019 | 2020 | 2021 | SEM |
|---|---|---|---|---|---|
| DM g.kg$^{-1}$ | 364 ± 30 [b] | 411 ± 55 [c] | 339 ± 14 [a] | 369 ± 20 [b] | 3.0 |
| Starch g.kg$^{-1}$ DM | 233 ± 44 [a] | 321 ± 28 [b] | 310 ± 27 [b] | 313 ± 24 [b] | 4.6 |
| aNDF g.kg$^{-1}$ DM | 499 ± 29 [c] | 443 ± 27 [b] | 435 ± 24 [b] | 415 ± 34 [a] | 3.8 |
| CP g.kg$^{-1}$ DM | 85.7 ± 7.6 [d] | 82.5 ± 5.4 [c] | 78.2 ± 6.7 [b] | 72.2 ± 4.8 [a] | 0.7 |
| Ash g.kg$^{-1}$ DM | 40.6 ± 4.7 [b] | 36.5 ± 5.5 [a] | 37.2 ± 5.5 [a] | 38.3 ± 2.9 [a] | 0.5 |
| aNDFD % | 56.1 ± 5.2 [b] | 49.4 ± 8.3 [a] | 53.3 ± 6.2 [b] | 49.4 ± 6.0 [a] | 0.8 |
| NEL MJ.kg$^{-1}$ DM | 6.30 ± 0.20 | 6.31 ± 0.21 | 6.35 ± 0.14 | 6.32 ± 0.20 | 0.02 |
| MPP kg.t$^{-1}$ DM | 1988 ± 63 | 1990 ± 67 | 2002 ± 43 | 1995 ± 62 | 7.62 |
| BMP L.kg$^{-1}$ DM | 387 ± 8 [c] | 364 ± 10 [b] | 365 ± 10 [b] | 351 ± 8 [a] | 1.26 |

DM = dry matter; aNDF = amylase-treated neutral detergent fibre; CP = crude protein; aNDFD = aNDF digestibility; NEL = net energy of lactation; MPP = milk production prediction; BMP = biochemical methane potential; ± = standard deviation; [a,b,c,d] means with different superscripts differ significantly ($p < 0.05$).

Agriculture is vulnerable to climate change at the global level. Climate change is a major concern and has adverse impacts on food production, food quality and food security [13]. The growing season temperature record (Table 2) was higher than the normal long-term weather range from 1981 to 2010 at both sites, L1 and L2. The growing season temperature trends were similar at both sites, but with the difference that temperatures were lower at location L1 than at L2 each year, even within the normal long-term weather range from 1981 to 2010. Growing season temperatures were higher in both locations during the drier years of 2018 and 2019 than in 2020 and 2021. The fact that 2021 was an exceptionally cool spring (April and May) may have influenced the results.

While L2 has always been warmer than L1, less precipitation has fallen there (Table 3), even in the normal long-term weather range from 1981 to 2010. For the first two years of the experiments (2018 and 2019), extremely little precipitation fell at both sites, and even less at L2 than L1. Compared to the normal long-term weather range of 1981 to 2010, very little precipitation fell in April in all four years at both sites. It turned out that each year, the weather pattern had a slightly different effect on the chemical composition of the hybrids (Table 5). In 2018, there was significantly lower starch content but higher aNDF and CP content. In 2019, there was significantly lower ash content but higher DM and CP content. In 2020, there was significantly lower DM and ash content but higher starch, aNDF and CP content. In 2021, there was a significantly lower aNDF, CP and ash content but higher DM and starch content. In 2018 and 2019, years with higher average temperature and lower rainfall, the CP content was significantly higher when compared to 2020 and 2021. The influence of climatic events, i.e., mainly drought, is highlighted by, e.g., Golbashy et al. [30] and Kumar et al. [31].

The fact that weather also influences other indicators is confirmed by several scientific papers. Hussain et al. [32] found that, although there was a poor correlation between the FAO group and the percentage of borer damage, a stronger positive correlation was found between the percentage of damage and the average air temperature in June, ranging from 20 °C to 24.5 °C, and was negatively correlated with relative humidity, ranging from 50% to 80%.

Maize is a sensitive crop to drought and heat stress, especially if it occurs unevenly during the reproductive stages of development [33]. The importance of the interaction of weather patterns during the growing season and the characteristics of the growing site has been emphasised by, e.g., Commission Regulation [20]. Although at first glance the growing conditions for maize were very similar at both L1 and L2, there were differences between them that affected both the results of chemical analyses and nutritional quality indicators. The only factor that did not differ between the two sites was BMP. In the plot in Prague (L1), the DM, starch, NDF and ash contents were statistically lower than in Troubsko (L2). Conversely, the aNDFD, NEL and MPP contents were statistically higher (Table 6). The differences in climatic conditions for each year are shown in Tables 2 and 3. In L1, temperatures were lower and precipitation was higher than in L2 (and also in the

normal long-term weather range) in all years during the growing season. According to Bažok et al. [33], the co-occurrence of drought and heat is more severe for maize growth than a single stress.

**Table 6.** Chemical composition and feed value of hybrids by location (L).

| Items | L1 | L2 | SEM |
|---|---|---|---|
| DM g.kg$^{-1}$ | 367 ± 24 [a] | 374 ± 55 [b] | 2.1 |
| Starch g.kg$^{-1}$ DM | 299 ± 37 [a] | 289 ± 56 [b] | 3.3 |
| aNDF g.kg$^{-1}$ DM | 440 ± 39 [a] | 456 ± 44 [b] | 2.7 |
| CP g.kg$^{-1}$ DM | 79.9 ± 9.8 | 79.5 ± 5.6 | 0.5 |
| Ash g.kg$^{-1}$ DM | 35.0 ± 2.9 [a] | 41.3 ± 4.6 [b] | 0.4 |
| aNDFD % | 54.3 ± 5.5 [b] | 49.8 ± 7.6 [a] | 0.55 |
| NEL MJ.kg$^{-1}$ DM | 6.41 ± 0.15 [b] | 6.23 ± 0.18 [a] | 0.02 |
| MPP kg.t$^{-1}$ DM | 2022 ± 46 [b] | 1965 ± 57 [a] | 5.39 |
| BMP L.kg$^{-1}$ DM | 366 ± 16 | 367 ± 16 | 0.89 |

L1 = locality Prague; L2 = locality Troubsko; DM = dry matter; aNDF = amylase-treated neutral detergent fibre; CP = crude protein; aNDFD = aNDF digestibility; NEL = net energy of lactation; MPP = milk production prediction; BMP = biochemical methane potential; ± = standard deviation; [a,b] means with different superscripts differ significantly ($p < 0.05$).

### 3.3. The Influence of the Hybrid

The effects of the type of maize hybrids and their maturity stage on their yield and, for cattle, their nutritional characteristics have been presented by many authors [34–40].

Table 7 shows the main characteristics of the maize hybrids we tested. The hybrids are ranked according to the FAO number that characterises the expected maturation period, with the first two hybrids (H1 and H2) belonging to the 'early' hybrids and the remainder belonging to the 'early–medium' hybrids. The small difference in the earliness of the hybrids (maximum of 40 FAO degrees) was chosen so that they could be harvested at a uniform date, which is what happened in both locations in each year. The plan for the harvest date was to follow the recommendation that the optimum grade for silage corn was a two-thirds milk line with some flexibility between a quarter and two-thirds [20]. Harvest maturity at the two-thirds milk line of grain is also recommended by Marchesini et al. [1]. Their research showed that at the same harvest date, the earlier hybrids measured higher DM and were predicted to have better fermentation results when used for silage. When earlier hybrids are harvested at the same date as those with higher FAO numbers, it can be assumed that a lower yield will be obtained [41]. Their conclusions were obtained from 822 samples of fresh maize plants from hybrids of early and late classes harvested at three maturity stages (early, middle and late), for three consecutive years and in three locations with different soil fertility.

**Table 7.** Chemical composition and feed value of hybrids (H) irrespective of the year of cultivation.

| Items | H1 | H2 | H3 | H4 | SEM |
|---|---|---|---|---|---|
| DM g.kg$^{-1}$ | 370 ± 38 [ab] | 370 ± 46 [ab] | 379 ± 51 [b] | 364 ± 32 [a] | 3.0 |
| Starch g.kg$^{-1}$ DM | 304 ± 47 [b] | 283 ± 47 [a] | 284 ± 52 [a] | 304 ± 42 [b] | 4.6 |
| aNDF g.kg$^{-1}$ DM | 446 ± 42 [ab] | 454 ± 43 [b] | 455 ± 38 [b] | 437 ± 45 [a] | 3.8 |
| CP g.kg$^{-1}$ DM | 78.4 ± 8.7 [a] | 81.2 ± 6.9 [b] | 78.2 ± 8.4 [a] | 80.9 ± 7.7 [ab] | 0.7 |
| Ash g.kg$^{-1}$ DM | 38.5 ± 4.9 [b] | 36.3 ± 4.4 [a] | 39.4 ± 6.0 [b] | 38.4 ± 4.1 [b] | 0.5 |
| aNDFD % | 49.3 ± 6.5 [a] | 51.0 ± 7.0 [ab] | 54.4 ± 6.7 [c] | 53.6 ± 6.9 [bc] | 0.78 |
| NEL MJ.kg$^{-1}$ DM | 6.24 ± 0.19 [a] | 6.29 ± 0.19 [ab] | 6.36 ± 0.20 [bc] | 6.38 ± 0.13 [c] | 0.02 |
| MPP kg.t$^{-1}$ DM | 1970 ± 59 [a] | 1985 ± 61 [ab] | 2007 ± 64 [bc] | 2013 ± 42 [c] | 7.62 |
| BMP L.kg$^{-1}$ DM | 370 ± 17 [b] | 366 ± 15 [ab] | 368 ± 15 [b] | 363 ± 17 [a] | 1.26 |

DM = dry matter; aNDF = amylase-treated neutral detergent fibre; CP = crude protein; aNDFD = NDF digestibility; NEL = net energy of lactation; MPP = milk production prediction; BMP = biochemical methane potential; ± = standard deviation; [a,b,c] means with different superscripts differ significantly ($p < 0.05$).

The largest difference in DM at harvest, 15 g.kg$^{-1}$, was between H3 and H4. This difference was between hybrids with different senescence patterns, with H3 being the only hybrid tested to have standard senescence (NSG). The other hybrids tended to be 'stay-green' (SG), which is a term used to describe genotypes that have delayed leaf senescence compared to reference genotypes [42]. This was also reflected in the highest aNDF content and aNDFD digestibility (Table 7). In addition, it is worth noting that SG hybrids are advantageous to grow because they have a 'wider harvest window', i.e., they can be harvested over a longer period without significant changes in the DM content and chemical composition of whole maize plants.

For the same type of hybrid (i.e., stay-green), the stage of grain maturity and the whole plant DM is closely related. When comparing the different types of hybrids (e.g., stay-green and dry down), this relationship may be different. Differences in DM content, nutrients and digestibility are also given by proportion grain (ear) to the other parts of the plant [43].

Hybrids for testing were chosen so that the differences between them were not only in earliness. The difference between hybrids was also in grain type (H2 and H3 semi-flint, H1 and H4 turning to flint [44]). However, this parameter probably did not have a significant effect on the results. The mode of ripening (senescence) may have had a greater effect; only H3 ripened uniformly, with the other hybrids having slight SG traits.

According to chemical composition, hybrid H1 had significantly lower CP and higher starch and ash content; hybrid H2 had significantly lower starch and ash content and higher aNDF and CP; and hybrid H3 had significantly lower starch and CP content and higher DM, aNDF and ash content. By maturity stage, the medium–late hybrid H3 had significantly lower starch and CP content and higher DM, aNDF and ash content.

The digestibility of aNDFD was lowest for H1, which was also reflected in NEL and MPP. This agrees with the statement of Jimenez et al. [45] that due to higher NDF digestibility, maize tends to have higher NEL and a higher estimate of potential milk production per hectare and per ton DM (MPP), determined using the MILK 2006 program according to Shaver et al. [8]. These indices (aNDFD, NEL and MPP) were higher in the later hybrids (H3 and H4). Hybrids H2 and H4 are intended not only for silage but also for use in biogas plants. However, this is not evident from the results; on the contrary, hybrid H4 had the lowest gas production. Agricultural practice requires more detailed and accurate research data and information. For farmers, choosing the right maize hybrid is crucial. Sometimes, however, under practical conditions, some characteristics of hybrids as declared by their sellers are not always apparent. In particular, the biogas maize ideotype is very difficult to find among current hybrids [46].

*3.4. Correlation between Nutrient Characteristics of Hybrids and Factors That May Have Influenced the Evaluation of Hybrids*

Table 8 shows the correlations by year, location and hybrid characteristics. Relationships were analysed using the Pearson's correlation coefficient (r, 1 or −1 depending on whether the variables were positively or negatively related [47]). The r coefficient values for correlation were interpreted according to Prior and Haerling [48]: very strong correlation (±0.91 to ±1.00); strong correlation (±0.68 to ±0.90); moderate correlation (±0.36 to ±0.67); weak correlation (±0.21 to ±0.35); and negligible correlation (0 to ±0.20). The colder and wetter the year, the higher the starch content (positively, moderately related) and the lower the aNDF and CP content, respectively, and the lower the BMP (negatively, strongly related). The cooler and wetter the site, the lower the DM and the higher the ash content but the lower the NEL and BMP. The BMP was not affected by location. The later the hybrid, the higher its energy value and MPP. While the correlations by years, locations or hybrids were significant mainly at the level of significance $p < 0.001$, the correlations by hybrid character were significant at $p < 0.01$. The starch content only slightly positively increased with the trend of being endosperm flint while being NSG. The SG hybrids had lower aNDF but higher aNDFD. The aNDFD, NEL and MPP values only weakly positively increased along with the trend of being endosperm flint and being NSG. This again supports the

conclusions made by Herrmann and Rath [46] that higher aNDFD tends to result in higher NEL and MPP in maize.

**Table 8.** Correlation by years, locations and characteristics of hybrids.

| | | Factors | | | Characteristics of Hybrids | | |
|---|---|---|---|---|---|---|---|
| **Items** | **Years** | **Locations** | **Hybrids** | **Maturity** | **Endosperm** | **Senescence** | **Use for** |
| DM g.kg$^{-1}$ | −0.03 | −0.40 ** | 0.01 | 0.03 | −0.09 | −0.10 | 0.09 |
| Starch g.kg$^{-1}$ DM | 0.63 ** | −0.23 * | 0.19 | 0.01 | 0.22 * | 0.23 * | 0.00 |
| aNDF g.kg$^{-1}$ DM | −0.75 ** | 0.26 * | −0.10 | −0.05 | −0.15 | −0.21 * | 0.06 |
| CP g.kg$^{-1}$ DM | −0.66 ** | −0.04 | −0.11 | −0.01 | 0.00 | −0.05 | −0.17 |
| Ash g.kg$^{-1}$ DM | −0.16 | 0.76 ** | 0.14 | 0.15 | 0.06 | 0.03 | 0.16 |
| aNDFD % | −0.26 * | −0.32 ** | 0.26 | 0.09 | 0.28 * | 0.25 * | 0.03 |
| NEL MJ.kg$^{-1}$ DM | 0.06 | −0.48 ** | 0.29 ** | 0.04 | 0.28 * | 0.28 * | 0.09 |
| MPP kg.t$^{-1}$ DM | 0.06 | −0.48 ** | 0.29 ** | 0.04 | 0.28 * | 0.28 * | 0.09 |
| BMP L.kg$^{-1}$ DM | −0.76 ** | 0.04 | −0.13 | 0.02 | −0.08 | −0.16 | −0.15 |

DM = dry matter; aNDF = amylase-treated neutral detergent fibre; CP = crude protein; aNDFD = aNDF digestibility; NEL = net energy of lactation; MPP = milk production prediction; BMP = biochemical methane potential; significant at * ($p < 0.01$), ** ($p < 0.001$).

## 4. Conclusions

It was confirmed that factors such as weather during the growing season, habitat characteristics and the characteristics of the maize hybrids themselves influenced the chemical analysis of fresh maize samples and their use in cattle feed and biogas plants. In our experiment, NEL and MPP values were lowest in the early hybrid DKC3872 with FAO 240–250 and highest in the medium early hybrid DKC3575 with FAO 270–280. On the other hand, BMP was highest in DKC3872 and lowest in DKC3575. Detailed analyses of the individual factors can be an important basis for agronomists to decide which hybrids to sow, how to treat them during the growing season according to weather patterns or how to harvest them to make the best use of their nutritional properties, including the ability to produce milk or biogas. In fact, it was found that the chemical analysis of fresh maize can also predict indicators such as aNDFD, NEL, MPP and BMP. Such information is then very useful for zootechnicians and biogas plant operators. For decades, maize has been bred for human and livestock nutritional and industrial purposes, but not for biogas production. While high methane yields can be achieved using various breeding strategies, the biogas maize ideotype is still hard to find among current hybrids. Therefore, more studies are needed to evaluate the effect of the chemical composition of maize hybrids on methane production. This study is preliminary and must be repeated with more hybrids and on more different conditions.

**Author Contributions:** Conceptualisation, R.L., F.J., V.J. and J.N.; methodology, R.L., V.J. and J.L.; formal analysis of data, P.H.; performing experiments, V.J., Y.T., A.V., P.K., V.K., M.G. and J.L.; writing—original draft preparation, R.L. and F.J.; supervision, P.H. and J.N. All authors have read and agreed to the published version of the manuscript.

**Funding:** This research was funded by the Ministry of Agriculture grant number QK1810137 and MZE-RO0718.

**Institutional Review Board Statement:** The protocol of the present experiment was approved by the Animal Care and Use Committee, Institute of Animal Science, Prague, Czech Republic (Act No. 359/2012 Coll.).

**Informed Consent Statement:** Not applicable.

**Data Availability Statement:** Data are available on request due to restrictions, e.g., privacy or ethical. The data presented in this study are available on request from the corresponding author. The data are not publicly available due to privacy.

**Conflicts of Interest:** The authors declare no conflict of interest.

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
