# Peer review of "Pilot Study on Predictive Traits of Fresh Maize Hybrids for Estimating Milk and Biogas Production"

_agriculture, doi:10.3390/agriculture12040559_

Round 1

Reviewer 1 Report

Dear Authors,

The paper entitled „Predictive traits of fresh maize hybrids for estimating milk and biogas production” focused on evaluation selected predictive traits of fresh maize hybrids based on chemical analyses using a multivariate approach to estimate their use in silage production for ruminant feeding, and possibly for methane production. Authors used MILK 2006 system for MPP evaluation and BMP for methane potential assessment. The obtained results are interesting from cognitive and practical point of view.

But at the beginning I have some doubts about the sense of evaluation MPP and BMP in fresh maize due to the fact that corn silage is used as forage (not fresh maize) in ruminants feeding. Also silage is probably more often used as a substrate for biogas production. So, I would like to ask the Authors for their opinion on these issues. Maybe it would be good to explain this in the introduction.

Next, MPP was calculated according to MILK 2006, originally developed to evaluate corn silage.  In the present study, MPP was calculated for fresh maize. During the maize conservation process, DM losses occur. Are you not afraid that the results obtained for MPP may not be completely reliable? What is your opinion on this issue?

Biochemical methane potential (BMP) tests are a technique to determine the methane potential and biodegradability of wastewater and waste biomass. Are there any recommendations for using this method to assess BMP for fresh maize too?

The manuscript is good written, but it needs revisions as reported in the specific comments.

In Abstract I suggest adding a one-two sentence introduction to the subject of study. What is the problem, what is the subject of study?

Introduction. I propose, basing on the literature, to add some information on the biogas production from maize. Please provide the information about MPP and BMP, about the assumptions of these methods and recommendations for their use in practice. Next what is the novelty of your research? The necessity and innovation of the article should be presented in the last paragraph of introduction section.

Statistical Analysis: Please specify the experimental design, and specify the experimental factors?

Table 2, 5, 6. The content of chemical components and feed value parameters are expressed in % DM, MJ/kg, kg/t and l/kg. I suggest using units that are compatible with the International System of Units: g·kg-1, MJ·kg-1, kg·t-1 and l·kg-1.

Table 3 and 4 should be moved to MM section and the course of weather conditions in each growing season and each location should be discussed in details. 

Author Response

Response to review of the paper entitled: „ Pilot study on predictive traits of fresh maize hybrids for es-timating milk and biogas production “ (Title was corrected based on reviewer suggestion.).

Dear reviewer,

Thank you very much for your inspiring suggestions. We accepted all of them and tried to incorporate them into the text. Because we moved some tables, we changed their numbers. Because we added some references, we changed their numbering.

Kind Regards

Authors

Your comment number 1:

But at the beginning I have some doubts about the sense of evaluation MPP and BMP in fresh maize due to the fact that corn silage is used as forage (not fresh maize) in ruminants feeding. Also silage is probably more often used as a substrate for biogas production. So, I would like to ask the Authors for their opinion on these issues. Maybe it would be good to explain this in the introduction.

Response 1:

Many thanks. Your point is well taken. But …

The present study compares several hybrids and the effects on their quality when tested under the same conditions. Theoretical (potential) values (better vs. worse) are determined for them, not absolute ones. So, the main idea is to compare hybrids with each other. That's what farmers want – a clear and simple statement that they understand. (L54 to 62)

Your comment number 2:

Next, MPP was calculated according to MILK 2006, originally developed to evaluate corn silage.  In the present study, MPP was calculated for fresh maize. During the maize conservation process, DM losses occur. Are you not afraid that the results obtained for MPP may not be completely reliable? What is your opinion on this issue?

Response 2:

Again, the main idea is to compare hybrids with each other.

We are far from the first to use MILK 2006 on fresh plants, such as Mussadiq et al. (2013) or Moreno-Resendez et al. (2017). We have added those references. (L60)

Because comparative experiments with maize silage and animals are difficult and expensive, we tried to replace them with comparative experiments with fresh plants, recognizing a certain error in the difference between silage and fresh plant, because during the maize conservation process DM losses occur. Yes, dry matter is lost during the maize silage fermentation process, but the losses depend on many other factors; some absolute value cannot be used. In addition, it should be noted that even the results of MILK 2006 may not be completely reliable, the system was developed mainly using the digestibility of NDF after incubation for 48 hours. At the same time, the feed passes through the digestive system of ruminants for about 24 hours and a lot depends on the nature of the mixed feed ration. The aim of the 48-hour incubation is therefore to make full use of the potential of the plants (silage) in order to minimize the variance of values; in that case, according to MILK 2006, the MPP is actually a little overrated.

Your comment number 3:

In Abstract I suggest adding a one-two sentence introduction to the subject of study. What is the problem, what is the subject of study?

Response 3:

OK, we have added:

Farmers need information on which maize hybrid is better than the others and under what conditions. They demand that this information be clear, simple and understood.

Using these indicators from fresh maize, information on differences between hybrids can be effectively obtained, albeit with some of the shortcomings of this proposed method. L14 to 19)

Your comment number 4:

Biochemical methane potential (BMP) tests are a technique to determine the methane potential and biodegradability of wastewater and waste biomass. Are there any recommendations for using this method to assess BMP for fresh maize too?

Response 4:

The evaluation of fresh maize as a substrate for biogas production has been discussed in detail by Weißbach [17], Weißbach [18] or Amon et al. [19].

Your comment number 5:

Introduction. I propose, basing on the literature, to add some information on the biogas production from maize. Please provide the information about MPP and BMP, about the assumptions of these methods and recommendations for their use in practice. Next what is the novelty of your research? The necessity and innovation of the article should be presented in the last paragraph of introduction section.

Response 5:

OK, we have added:

Fresh maize as a material for the study was chosen with the knowledge of a certain error in comparison with silage (dry matter losses occur during fermentation). But farmers are mainly interested in the differences between hybrids, respectively differences between factors (ie which hybrid is better than the others). Absolute values are less important in this case. It is also important that experiments with silage or even animals are difficult and expensive. Evaluation of maize hybrids according to MILK 2006 in fresh plants was used, for example, by Mussadiq et al. [15] or Moreno-Resendez et al. [16]. (L59 to 62)

Your comment number 7:

Statistical Analysis: Please specify the experimental design, and specify the experimental factors?

Response 7:

OK, we have done it. (L133 to 144)

Your comment number 8:

Table 2, 5, 6. The content of chemical components and feed value parameters are expressed in % DM, MJ/kg, kg/t and l/kg. I suggest using units that are compatible with the International System of Units: g·kg-1, MJ·kg-1, kg·t-1 and l·kg-1.

Response 8:

OK, we have done it.

Your comment number 9:

Table 3 and 4 should be moved to MM section and the course of weather conditions in each growing season and each location should be discussed in details.

Response 9:

OK, we have done it. (L95 to 104; L180 to 224)

Reviewer 2 Report

The study discusses predictive traits of fresh maize hybrids for estimating milk and biogas production. Interesting topic however I believe that the numerosity of the selected hybrids and the variability between hybrids is not enough to study the topic. In fact, I suggest adding to the title: Pilot study on predictive traits… Moreover, I suggest also adding around the text, and mainly in the conclusions, that this study is very preliminary and must be repeated with more hybrids and on more different conditions.

Please provide more information about statistical analysis. It is not clear how the multivariate model was constructed after Tukey’s test was applied. In fact, in the results table 2, 5, and 6 seems to result from a Tukey’s test, results reported before the multivariate analysis (table 8 I suppose). Moreover, it is not reported the type of correlation and the importance of it (please see: doi:10.3390/ani10081334 to check how to discuss the correlation results).

Finally, please report the formula of aNDFD, NEL, MPP, and BMP calculations.

Author Response

Response to review of the paper entitled: „ Pilot study on predictive traits of fresh maize hybrids for es-timating milk and biogas production “ (Title was corrected based on reviewer suggestion.).

Dear reviewer,

Thank you very much for your inspiring suggestions. We accepted all of them and tried to incorporate them into the text. Because we moved some tables, we changed their numbers. Because we added some references, we changed their numbering.

Kind Regards

Authors

Your comment number 1:

The study discusses predictive traits of fresh maize hybrids for estimating milk and biogas production. Interesting topic however I believe that the numerosity of the selected hybrids and the variability between hybrids is not enough to study the topic. In fact, I suggest adding to the title: Pilot study on predictive traits… Moreover, I suggest also adding around the text, and mainly in the conclusions, that this study is very preliminary and must be repeated with more hybrids and on more different conditions.

Response 1:

Yes, both text editing proposals were accepted. Corrections are incorporated into the paper. (L2; L26 and 27; L340 and 341)

Your comment number 2:

Please provide more information about statistical analysis. It is not clear how the multivariate model was constructed after Tukey’s test was applied. In fact, in the results table 2, 5, and 6 seems to result from a Tukey’s test, results reported before the multivariate analysis (table 8 I suppose). Moreover, it is not reported the type of correlation and the importance of it (please see: doi:10.3390/ani10081334 to check how to discuss the correlation results).

Response 2:

Yes, both proposals were accepted. Corrections are incorporated into the paper. (L133 to 144)

Your comment number 3:

Finally, please report the formula of aNDFD, NEL, MPP, and BMP calculations.

Response 3:

Formulas of aNDFD and NEL were inserted to Material and Methods. (L114 and L117 to 119)

For MPP was used “program” MILK 2006” where values of DM, CP, NDF, starch, EE and ash contents and aNDFD values were inserted. Then MPP were from these values automatically calculated. Unfortunately, specific equations are not available from this program. That is why we cannot give a specific calculation. However, a citation from the program's source and authors is provided. (L119 to 121)

The BMP values were not detected by calculations but they were measured by laboratory method. This method was described in detail in Material and Methods. (L121 to 128)

Round 2

Reviewer 1 Report

Dear Authors,

the text of manuscript was improved according to my suggestion and can be accepted for publication in present form. 

Reviewer 2 Report

The paper improved a lot, I do not have any more concerns.

This manuscript is a resubmission of an earlier submission. The following is a list of the peer review reports and author responses from that submission.

Round 1

Reviewer 1 Report

The scientific goal of this paper is rather unclear for me. In general, I see in it a large amount of data recorded by the NIR technique, and an observation of certain facts. I do not know what this data would show and what scientific significance the observations have. In my opinion, the article should be thoroughly rewritten taking this point of view into account. 

Reviewer 2 Report

I strongly recommend authors read once more through the article before resubmitting. Many sentences are incomplete. Please AND please! read once more. For example, kindly have a look at the abstract. it's poorly written and the last sentence is incomplete. Although there are scientific merits, I suggest rewriting or language revision before revising for the scientific content. Its sometimes hard to judge the author's intentions, since expressions are not easy to understand (or not scientific and suitable for a high-quality journal).